# LC-MS/MS Validation and Quantification of Cyanotoxins in Algal Food Supplements from the Belgium Market and Their Molecular Origins

**DOI:** 10.3390/toxins14080513

**Published:** 2022-07-27

**Authors:** Wannes Hugo R. Van Hassel, Anne-Catherine Ahn, Bart Huybrechts, Julien Masquelier, Annick Wilmotte, Mirjana Andjelkovic

**Affiliations:** 1Unit Toxins, Organic Contaminants and Additives, Sciensano, Rue Juliette Wytsmanstraat 14, 1050 Brussels, Belgium; bart.huybrechts@sciensano.be (B.H.); julien.masquelier@sciensano.be (J.M.); 2InBios-Centre for Protein Engineering, Departement of Life Sciences, University of Liège, Allée du six Août 11, 4000 Liège, Belgium; awilmotte@uliege.be; 3BCCM/ULC Collection, University of Liège, Allée du six Août 11, 4000 Liège, Belgium; annecatherine.ahn@gmail.com; 4Risk and -Health Impact Assessment, Sciensano, Rue Juliette Wytsmanstraat 14, 1050 Brussels, Belgium; mirjana.andjelkovic@sciensano.be

**Keywords:** ultra-high performance liquid chromatography, Tandem Mass Spectrometry, BGAS, microcystin, *Microcystis*, sequencing, food safety

## Abstract

Food supplements are gaining popularity worldwide. However, harmful natural compounds can contaminate these products. In the case of algae-based products, the presence of toxin-producing cyanobacteria may cause health risks. However, data about the prevalence of algal food supplements on the Belgian market and possible contaminations with cyanotoxins are scarce. Therefore, we optimized and validated a method based on Ultra High Performance Liquid Chromatography-Tandem Mass Spectrometry to quantify eight microcystin congeners and nodularin in algal food supplements. Our analytical method was successfully validated and applied on 35 food supplement samples. Nine out of these samples contained microcystin congeners, of which three exceeded 1 µg g^−1^, a previously proposed guideline value. Additionally, the *mcyE* gene was amplified and sequenced in ten products to identify the taxon responsible for the toxin production. For seven out of these ten samples, the *mcyE* gene could be amplified and associated to *Microcystis* sp. EFSA and posology consumption data for algal-based food supplements were both combined with our toxin prevalence data to establish different toxin exposure scenarios to assess health risks and propose new guideline values.

## 1. Introduction

The use of cyanobacteria- and *Chlorella*-based food supplements are becoming more and more common worldwide. Overall consumption of food supplements in the United States of America increased by 12% compared to a decade earlier as shown by the Council for Responsible Nutrition (CRN) Consumer Survey in 2017 [1]. The survey further revealed that the supplements are primarily consumed to improve overall health and energy [1]. However, similar consumption data are not yet available for Europe nor Belgium, which prevents a more realistic assessment of a health risk. Following the last National Consumption Survey in Belgium [2], food supplements on the basis of ‘spirulina’ and *Chlorella* were the third most consumed supplements, not belonging to the group of vitamins and minerals. They were ranked below yeast supplements and bee products. However, these results were obtained with food frequency surveys, which are only descriptive. A risk assessment for cyanobacteria- and *Chlorella*-based food supplements is necessary as a more frequent ingestion of these products could also lead to an increased exposure of the public to the ingredients used for their production, which might unintentionally contain harmful compounds. For instance, natural toxins such as phyto-, myco-, or phycotoxins can sometimes be found in these types of food supplements [3,4].

Two types of blue-green algae-based food supplements (BGAS) are commonly found on the market. The first product is ‘spirulina’, which is mainly produced from cultivated *Arthrospira platensis* and occasionally *Arthrospira maxima*. The denomination ‘spirulina’ is a commercial one. It originates from the time before molecular taxonomy methods proved that the true *Spirulina* sp. were very different from the cyanobacteria of the *Arthrospira* genus, used as food supplement [5]. In this paper, any *Arthrospira* sp. based product will be referred under the general product name ‘spirulina’. The cultivation of these cyanobacteria is generally performed in open air ponds and occasionally in closed incubators [6,7,8,9]. The culture medium required for growth of *Arthrospira* sp. has a high pH and high salinity, which protects against contamination by other cyanobacteria [9]. The second BGAS product found on the market is based on the cyanobacterial species *Aphanizomenon flos-aquae*. The product names most associated to this food supplement are Klamath and AFA (*Aphanizomenon flos-aquae)*. *Aphanizomenon flos-aquae* occurs naturally in the Klamath Lake in Oregon, USA, and is being harvested there for already more than 30 years [10]. However, the harvest of naturally occurring blooms is not unique to Oregon. In South America and Africa, local populations have been harvesting *Arthrospira* sp. from soda lakes for centuries [11].

Natural lakes usually contain mixed populations of cyanobacteria. By harvesting natural cyanobacteria, potentially toxic organisms could be included in the BGAS. Moreover, food supplements can also be contaminated during the after-harvest processing. The presence of toxins produced by cyanobacteria was already shown in commercially available BGAS by several studies [3,12,13,14,15,16,17].

Different types of cyanotoxins exist based on their effect on the human body, and include neurotoxins and hepatotoxins. Neurotoxins consist of three major members: saxitoxin (STX), anatoxin-a (ATX), and β-methylamino-L-alanine (BMAA). STX is regularly monitored and might be found in shellfish (filter feeding organisms) but until now, it was not reported in BGAS [18]. In contrast, ATX and BMAA were already detected in BGAS either by HPLC-LFD or HPLC-MS/MS [3,12,13,15].

Another major group of cyanotoxins are hepatotoxins, including cylindrospermopsin, nodularin and microcystin. The first two were never detected in BGAs [12,13,16,19,20,21]. This is in strong contrast with the results obtained for the microcystin congeners (MCs). These heptacyclic peptides with their specific ADDA ((2S,3S,8S,9S)-3-amino-9-methoxy-2,6,8-trimethyl-10-phenyl-4,6-decadienoic acid)-moiety are produced by different species of cyanobacteria (e.g: *Microcystis, Anabaena, Oscillatoria, Nostoc)* [22], and are commonly found in the environment as well as in BGAS [12,13,14,17,19,20,21,23,24,25,26,27].

The toxicity of microcystin and nodularin results from the interaction of their ADDA tail with protein phosphatase 1 (PPI) or 2a (PPIIa), enabling the covalently binding of the hepatotoxin to the enzyme and inhibiting the protein activity. In particular, the toxicology of MCs is well studied. They are primarily transported to the liver by bile salts, where specific organic anion transporting polypeptides (OATPS) transport the MCs into the cells [28,29,30,31]. The function of the liver cells is disrupted, causing liver cirrhosis after prolonged exposure [32,33,34,35]. Based on the available toxicological research, the World Health Organization (WHO) suggested a tolerable daily intake (TDI) for MC-LR of 0.04 µg kg _bw_^−1^ day^−1^. However, this does not take into account that multiple MCs can be present [36]. In addition, the structural variation between different congeners causes differences in chemical characteristics that can alter the toxicity of a congener [25]. Amino acids composition of the MCs at position 2 and 4 can be variable, while other positions can be methylated, demethylated and/or acetylated [25,37,38,39,40,41,42,43,44,45,46,47]. Overall, 279 different congeners have been reported [25].

Several analytical methods have already been used to detect MCs in BGAS. The ELISA method was first used to detect MCs in BGAS originating from the Upper Klamath lakes. Concentrations of MCs reaching up to 16.4 µg g^−1^ were found in 85 of the 87 tested products. MC-LR was determined by HPLC-separation and ELISA quantification as the most dominant congener in the products [14]. Other approaches, LC-MS, LC-MS/MS and Phospho-Inhibition Assay approach have been compared and used to determine MCs concentrations between 0.1 and 35.7 µg g^−1^ [19].

During the past years, the LC-MS(/MS) has evolved to become a commonly used technique to detect MCs in BGAS. Multiple groups have developed and validated methods capable of accurately detecting MCs. The amount of different congeners that they were able to detect, was primarily dependent on the availability of standards [12,13,17,20,21,23,48].

Furthermore, molecular techniques allow to screen for cyanotoxin synthetase genes and thus to determine the capacity to produce cyanotoxins in a given sample. The gene clusters are identified based on the *mcyE* and the *ndaF* genes encoding a microcystin and nodularin synthetase, respectively [49,50]. For example, Saker et al. [51] amplified the *mcyA* and *mcyE* genes in BGAS and corroborated the result with an ELISA test.

Currently, only a limited number of studies have evaluated the contamination of BGAS by cyanotoxins and even fewer have tried to quantify the health risk for the consumers based on consumption data. In this work, we explored the presence of eight MCs and NOD in BGAS sold on the Belgium market using a validated UHPLC-MS/MS method. In addition, the *mcyE* gene was amplified and sequenced from the positive samples to determine the primary producer of the toxin. Elucidating the presence and the source of the contamination allows a more comprehensive assessment of the public health risks. A new assessment of the daily exposure to MCs was made and new guideline values were proposed based on the available posology and consumption data for BGAS.

## 2. Results

### 2.1. Validation of UPLC-MS/MS Method

The validation of our method was successful for all nine toxins. The criteria for specificity were met. Namely, blank samples did not present toxin signals. All spiked samples showed signals for the quantifier and qualifier ion, and the ion ratio between both was within the limits set by the EU commission decision 2002/657/EC [52].

The LOQ for each toxin was determined as the lowest validated concentration (50 µg kg^−1^). Signal to Noise (S/N) values were above 10 for each toxin, shown in Table 1. Usually, LOD is calculated as 1/3 of the LOQ, which in this case would be ± 16.66 µg kg^−1^. However, the LOD was validated as 22.5 µg kg^−1^ because this was also the lowest point, for which S/N values were obtained. Table 1 shows that the S/N values for the LOD are above 3, as dictated by the guidelines EU commission decision 2002/657/EC [52].

Furthermore, the Mandel’s fitting test showed a preference for an exponential fit. However, the linear fits for all the curves showed R^2^ values higher or equal to 0.99, resulting in this fit being chosen to determine concentrations in the samples, as it is more straightforward to calculate (shown in Table 1) [53].

Additionally, a matrix effect was observed for seven out of nine toxins in the BGAS matrix during the validation, as shown by the non-parallel relationship between the two calibration curves (Figure 1) and further evaluated with a student *t*-test (Table 2). All congeners except MC-LA and NOD were affected by a matrix effect. The calibration curve was therefore made in blank matrix, which reduces the effect of the matrix on the quantification.

The apparent recoveries were calculated for the four different spiking levels (50 µg kg^−1^, 400 µg kg^−1^, 800 µg kg^−1^ and 1200 µg kg^−1^) and additionally as average of the four levels. Recoveries were situated between 69% and 104% (Table 1).

For the repeatability and reproducibility, the Horwitz ratio determined maximum values from 10.3 to 14.7% average variance and 15.5 to 22.0% coefficient of variation (CV), respectively. These thresholds were calculated for each toxin at each concentration level. The actual average variance and CV obtained from the measured concentrations were both below their respective threshold values for each toxin at each spiked concentration (Table 1).

At last, the measurement’s uncertainty for all toxins for all spike levels was comprised between 6.26% and 37.48% as seen in Table 1.

### 2.2. Toxin Occurrence in Food Supplements

Cyanotoxins, under the form of six MCs congeners, were detected in nine out of the 35 analyzed samples. Nodularin, MC-LW, and MC-LF were not detected in any of the samples. All contaminated products contained 100% *Aphanizomenon flos-aquae* with the exception of Mx-852, which was a mix of the ‘spirulina’, *Chlorella* sp. and *A. flos-aquae*. However, only MC-RR below LOQ was found in Mx-852 below LOQ. The other eight contaminated products contained total MCs concentration between 238.45 and 5645.33 µg kg^−1^, as shown in Table 3.

The congener compositions varied between the different products. Between two and six different congeners were detected in the eight MCs contaminated samples by our validated method (Figure 2). It should be noted that the samples with the highest congener diversity (Apha-430 and Apha-1230) contained MC-WR below LOQ, whereas the five other congeners could be quantified. MC-LA and MC-LR were the most prevalent toxins found in the products, as shown in Figure 2.

### 2.3. Microcystin Synthetase Encoding Genes

The HEP amplicons (385 bp) of seven food supplements (Apha-430, Apha-584, Apha-585, Apha-587, Apha-650, Apha862 and Apha-1230) corresponded to the sequences of the *mcy*E gene of different *Microcystis* sp. No HEP amplicons could be obtained from the samples Sp-475, Mx-582 and Apha-696. Moreover, the samples Apha-430, Apha-650, Apha-862 and Apha-1230 contained more than one *Microcystis* sp. strain as there were single nucleotide polymorphisms in their HEP amplicon sequence. Samples of Apha-584, Apha-585 and Apha-587 contained only a single HEP amplicon sequence, which was identical for the three samples and which could be associated with *Microcystis* sp. clone Bel-Nar12/07-1 (100% identity, accession: KF219536.1).

### 2.4. Exposure and Risk Assessment Estimate

Based on the EFSA Comprehensive Food Consumption database, the average and 95th percentile consumption from the acute consumption data of algae based formulations (e.g., ‘spirulina’, *Chlorella*) for all population groups (Adolescents, Adults, Elderly, Other children, Pregnant women, Vegetarians and Very elderly) is 2.6 g and 3.5 g, respectively. However, the data can be considered statistically robust only for the adults, with a number of observations higher than 60. Recalculating the consumption for adults, based on the data found in the EFSA database, provides values of 1.62 g and 2.58 g on average per day and at the 95th percentile per day, respectively. These values were considered during the toxin exposure estimations (estimated daily intake (EDI)) in Table 4.

Additionally, the posology data (suggested daily dose) found on the package of the tested products was also used as a measure of the consumption. The average suggested daily dose for each food supplement was 3.14 g day^−1^, while for the 95th percentile, this was 6.25 g day^−1^. Therefore, an average EDI was 0.02 µg kg_bw_^−1^ day^−1^ for an adult person of 70 kg (Table 4). In a worst case exposure scenario (only upper bound approach), where only the contaminated samples were included, the average EDI was 0.10 µg kg_bw_^−1^ day^−1^ and the 95th percentile exposure at the recommended doses was 0.33 µg kg_bw_^−1^ day^−1^. For the comparison, the exposure estimation was also done using the mean consumption calculated for the adults using the EFSA Food Consumption Comprehensive Database (Table 4).

In the worst case exposure scenario, the tolerable daily intake (TDI) [36] of 0.04 µg kg_bw_^−1^ for an adult was exceeded for both the average (0.10 µg kg_bw_^−1^ day^−1^) and 95th (0.33 µg kg_bw_^−1^ day^−1^) percentile EDI. For children, the TDI values would be severely exceeded by the EDI in both exposure scenario’s. This suggests that there might be a potential health concern risk for some consumers and in particular for children when contaminated products are consumed.

Furthermore, we suggest a maximum allowed MCs concentration in BGAS for adults, ranging from 0.70 to 1.08 µg MC-LR equivalent g ^−1^ day^−1^. The lower limit was calculated based on the 95th percentile of consumption based on posology data (representing high consumers in worst case exposure scenario. The upper limit represents the average consumer use (average consumption based on posology data). If children (15 kg average weight) might consume these BGAS, the maximal allowed level should be even lower (0.15 µg MC-LR equivalent g^−1^ day^−1^ BGAS). The average and 95th percentile of MCs concentration found in BGAS during our study for the worst case scenario were 1.84 µg MC-LR equivalent g^−1^ and 5.72 µg MC-LR equivalent g^−1^, respectively, and both surpass the suggested maximum allowed MCs concentration for adults and children.

## 3. Discussion

Based on the data from the literature cited in this study, it is clear that a ± 80% MeOH extraction is the most appropriate way to extract the MCs and NOD, with the addition of a mechanical extraction and/or sonication [12,17,20,21,48]. Moreover, no additional purification seems to be needed for BGAS samples for MS/MS analysis, which significantly reduces material cost and analysis duration.

Several quantification methods for MCs in food supplements have already been validated. Turner et al. (2018), validated a method using the same extraction method but a different detection and quantification method compared to our method, which uses a calibration curve in the matrix. A matrix-matched calibration curve should result in a more accurate determination of the recovery, as the matrix effect is taken into account. Our recoveries seem to be similar to the results from Turner et al. (2018), except for the lower recoveries of MC-LF. The values of the other parameters (LOD, LOQ, repeatability, …) are difficult to compare as different calculations were used [48]. Other validated methods are even more difficult to compare as the methods used different techniques during extraction and/or quantification of the methods. Parker et al. (2015) included a SPE step in their validated method. Up concentration of the samples during SPE lowers the LOD and LOQ, while optimizing recovery for all MCs and NOD and minimizing the matrix effect. However, this methodology does increase the needed time and the associated costs to perform the method [21]. Ortelli et al. (2008) validated a method for quantification of MC-RR, MC-YR, MC-LR, MC-LA and NOD using LC-MS. They used a time of flight MS instead of the triple quadrupole used in our method [20].

A more complex method for MCs quantification was also validated by Roy-Lachappelle et al. using LC-HRMS. The recovery and LOD/LOQ were comparable to our results, while the repeatability and reproducibility were lower [13].

During our screening of BGAS (*Chlorella* sp., ‘spirulina’ and Klamath derived products) on the Belgium food market, we found nine products that were contaminated with MCs with concentrations between 0.24 and 5.6 µg g^−1^ total microcystin, and with MC-LR and MC-LA as most abundant congeners. They were exclusively observed in products containing *Aphanizomenon flos-aquae*. Our concentrations fit perfectly with the earlier reported toxin concentrations ranging between 0 and 60 µg g^−1^ in BGAS using different methods (e.g., MS/MS, MS, ELISA, PPIA) [12,14,17,19,20,54,55]. Roy-Lachappelle et al. (2017) also found toxin concentrations between 0.25 and 8.2 µg g^−1^ total microcystin, present in ‘spirulina’ products, which is uncommon [13].

Besides UPLC-MS/MS methods, PCR amplification of the *mcyE* gene could be a valuable simple tool during screening of BGAS and detection of possible contaminations during the production process. The sequencing of the *mcyE* amplicon allowed to identify the producer of the microcystin. In seven MCs positive BGAS, the *mcyE* gene of *Microcystis* sp. could be detected by PCR [49]. In three other samples, PCR amplification failed, but this could be explained. The sample Sp-475 was a negative control made on a sample where no MCs were detected and the lack of amplification of the *mcyE* gene was expected. For sample Mx-582, MC-LR was detected but in too low concentration to be quantified. For sample Apha-696, a quantifiable level (499.52 total microcystin µg kg^−1^) of MCs was measured but it was the only Apha sample in tablet form. Therefore, we hypothesize that the treatment used for tablet production sheared the DNA too much for proper PCR amplification. The presence of *Microcystis* sp. in food supplements has previously be reported in several studies [17,51,56,57]. Therefore, we advise the producers to take additional precautions to avoid the presence of such known toxin-producing cyanobacteria and control their absence by regular screenings using microscopy or PCR.

However, knowing the presence of toxin-producing cyanobacteria or even the concentration of toxins is not sufficient to assess the risk to public health. However, a risk assessment for cyanotoxins in BGAS is difficult to perform, due to a lack of occurrence and toxicity data for some microcystin congeners, as well as a lack of regulation on food supplements, in particular regarding these toxins. For instance, the average daily consumption of BGAS should be known to calculate the risk. Yet, recommended doses are not uniform between different products. Generally, the amount of a tea spoon (2 g day^−1^) is recommended on a daily basis, although some products recommend a higher consumption or do not provide a recommended dose on the product packaging. Therefore, an average dose of 3.14 g day^−1^ was calculated on the basis of the posology data of the tested products. Additionally, there is a chance that users will not comply with the recommended dose. For instance, Dietrich and Hoeger [58] mentioned that parents might increase the dose for their children above the recommendation in an attempt to further increase the beneficial results of the BGAS but obtain the inverse effect. Marsan et al. found products with a recommended dose of up to 15 g day^−1^ [54]. Moreover, Gilroy et al. report recommended doses of BGAS up to 20 g day^−1^ [14]. If we use the data from the EFSA food consumption database, we calculated an average daily dose of 1.6 g of BGAS and 2.58 g day^−1^ in the highest 95th percentile of consumption for adults in the food category “Algae formula” (e.g., ‘spirulina’, *Chlorella*) [18]. This dose was two times lower than the average calculated in our study. Unfortunately, the consumption of these products in Belgium has not yet been quantified. The difference between recommended and real consumption complicates the evaluation and the recommendation of a guideline value.

The WHO considers a value of 0.04 µg kg_bw_^−1^ day^−1^ as a tolerable daily intake (TDI) for MCs, which is assumed to be a safe dose for a lifetime ingestion. Using the suggested doses, we have estimated that in a worst case exposure scenario (when MCs are present at the level found in the study), EDI might be on average as high as 0.1 µg kg_bw_^−1^ day ^−1^ for an adult person. However, for some consumers, this might be as high as 0.33 µg kg_bw_^−1^ day^−1^. These estimated values exceed the TDI value. Nevertheless, only a small proportion of the samples showed a contamination, and it was linked to the presence of *Aphanizomenon flos-aquae*. Gilroy et al. already proposed a regulatory value of 1 µg g^−1^ of BGAS consumption based on the TDI of MCs suggested by the WHO, which considers an average consumption of 2 g day^−1^ for an average human of 60 kg. Based on the results of our study, where the average consumption is above 2 g and the average body weight was estimated as 70 kg following the EFSA guidance, a similar recommendation can be given [59]. The MCs concentration should not exceed 1.08 µg g^−1^ BGAS for most consumers. Dietrich and Hoeger also mentioned that the supplements are also consumed by children, who, due to a lower body mass, are more susceptible to intoxication [58,60]. Recalculating a guideline value for infants and children would be advised, especially as BGAS consumption generally represents a long term exposure. If the same values for the consumption of BGAs would be applied to adults and children, the regulatory threshold would decrease more than four times to ensure a safe consumption for all the population below the TDI level. Based on the concentrations found in our study, the recommended regulatory level could be as low as 0.15 µg g^−1^. Similarly, the new WHO provisional guidelines for MCs in drinking water recommends a 3 µg L^−1^ total microcystin for children during short exposure. They also used the average body weight of a child (15 kg) while calculating the guideline value for recreational waters, instead of the average body weight of an adult (60 kg) [36].

For the Belgium market, three products exceeded the value proposed by Gilroy et al. Eight products contained a MC-LR equivalent concentration higher than 0.5 µg g^−1^. If the original consumption estimate from EFSA (3.5 g) was used, the guideline value for adults would become 0.8 µg g^−1^ MC-LR equivalent, meaning that sample Apha-862 would also be considered as unsafe.

Further research should be done to obtain more data concerning the daily consumption of BGAS as well as the toxin presence in the BGAS to improve the risk assessment for the consumers. A regular monitoring is also advisable, as Gilroy et al. showed that there could be variations in toxin concentration between different batches [14].

## 4. Conclusions

In this study, the validation of a quantitative UHPLC-MS/MS method to quantify eight MCs and NOD in BGAS is performed for the first time with a matrix-matched calibration curve. Validation results adhered to the pre-set parameters. Using this method, the presence of five MCs was determined in nine of the BGAS samples at concentrations that could pose a risk to the public health. Additionally, we were able to determine that the *mcyE* gene, involved in MCs production, originated from *Microcystis* sp. Although the recommended maximum MCs level in BGAS were calculated, the actual risk to the public health is difficult to determine. A lack of BGAS consumption data, MCs occurrence and toxicological data for most MCs needs to be resolved to obtain a proper risk assessment for the Belgian population. However, using the currently available data, multiple exposure scenarios were discussed and new guidelines were suggested. Establishing proper regulation and monitoring programs for MCs in BGAS would be advisable, especially for products harvested from the environment. These products are a protein rich and eco-friendly food source that should not be dismissed out of hand due to the possibility for toxin contamination, which is also possible in more ‘traditional’ food sources (e.g., phyto-, myco- and bacterial toxins).

## 5. Materials and Methods

### 5.1. BGAS Sampling on the Belgium Market

Samples were selected after an offline and online market inquiry. In total, 35 BGAS were selected with a balance between *Chlorella*, ‘spirulina’ and Klamath-based products and purchased within few weeks after the inquiry. Mixed products, with two or more of these ingredients were also included. More details on these samples are available in Table 3.

### 5.2. Reagents and Materials

The solvents for the extraction and for the basis of the mobile phase were UHPLC-MS grade. All toxin standards for MCs (MC-LR, MC-RR, MC-LA, MC-LY, MC-LF, MC-LW, MC-YR, MC-WR) and NOD came from Enzo Life Sciences^®^ (Antwerp, Belgium) and were received under the form of a solid powder. After dilution in 100% methanol (MeOH), mixed stock solutions were prepared in 50% methanol (MeOH) (50% Milli-Q water with 1% acetic acid (*v/v*)). The stock and the intermediate solutions were stored at −20 °C.

### 5.3. Sample Preparation and Extraction of Cyanotoxins

Before extraction, the tablets were crushed to a powder using a mortar. If the BGAS was enclosed in a capsule, it was opened and the powder was removed. The complete package content was processed to ensure homogeneity within the sample. Then, 0.5 g BGAS powder was weighed, and dissolved in 80% MeOH (*v/v*). This mixture was blended for 1 h in an overhead shaker. Thereafter, the sample was centrifuged for 10 min at 2500× *g*. The supernatant was removed and filtered through a Phenomenex 0.2 µm RC-syringe filter (Utrecht, The Netherlands).

### 5.4. Detection and Quantification of Cyanotoxins

To continue the analysis, each BGAS extract was diluted ten times. Samples were injected onto the UPLC-MS/MS system (Xevo TQ-S triple quadrupole mass spectrometer —Waters, Milford, MA, USA) at a volume of 5 µL. UHPLC and MS/MS conditions were already described in detail in earlier published works [27,61]. In short, during injection, the congeners were separated on a 1.7 µm, 2.1 mm × 100 mm BEH-C18 column from Waters © (Eten-Leur, The Netherlands) preceded by a 1.7 µm BEH C-18 VANGUARD pre-column from Waters©. A gradient elution with an initial high percentage (98%) water phase was used, while gradually increasing the proportion of acetonitrile over time.

The precursor, quantifier and qualifier ion used for detection can be found in Table 5, accompanied by corresponding collision energies and cone voltages.

Toxin concentrations were calculated by the TargetLynx extension included in the MassLynx V4.2 (Waters©) software based on dilution factors and a six-pointed calibration curve, which was made in blank matrix (sample extract containing no MCs or NOD) with concentrations between 0.5 µg L^−1^ (45 µg kg^−1^) and 50 µg L^−1^ (4500 µg kg^−1^). The calibration curve in the blank matrix corrects for the matrix effect. During each run, a quality control (QC) sample was also added to calculate the recovery of the toxins during the analysis run. A blank matrix sample was spiked with a mixture of all the toxin standards resulting in a final concentration of 800 µg kg^−1^ for each of the MCs and NOD. The QC was further extracted and analyzed in accordance with our method. Eventually, the observed concentration in the unknown (BGAS or Chlorella) samples should be corrected with the recovery.

### 5.5. Validation of the MS/MS Method

The validation study was performed using a blank (containing no MCs or NOD) ‘spirulina’ sample and completed before analysis of the BGAS samples. Aliquots of blank ‘spirulina’ powder were spiked with a toxin mix of eight MCs and NOD to validate the method. Each toxin was added at concentrations of 50 µg kg^−1^, 400 µg kg^−1^, 800 µg kg^−1^ and 1600 µg kg^−1^ in triplicate. The procedure was repeated on three different days. The following multiple parameters that are described below were evaluated to assess the validity of the method: specificity, limit of detection (LOD) and the limit of quantification (LOQ), apparent recovery, reproducibility, repeatability, measurement uncertainty, linearity and matrix effect.

The specificity of the method was stated as sufficient if the quantifier ion and the qualifier ion were both present during detection, and the ion ratio adhered to the EU commission decision’s 2002/657/EC limits [52]. Furthermore, no residual signal should be detected in the blank samples above 1% of the signal intensity found in the 800 µg kg^−1^ spike.

The LOQ was determined as the lowest concentration, for which a toxin was fully validated and the signal to noise (S/N) ratio was 10. The LOD was defined as 1/3 of the value of the LOQ if the S/N signal is higher than 3. Moreover, the boundaries set for the apparent recovery laid between 60% and 120% and the limits for reproducibility and repeatability were obtained using the Horwitz ratio. Reproducibility is calculated as the average variance of the validation results, representing the variability of the method over multiple days of analysis. Repeatability is calculated as the CV of the validation results representing variability of the method during one day of analysis. The measurement uncertainty was calculated as the double of the CV. Linearity of the standard curve was determined with a Mandel’s fitting test between concentrations of 0.1 and 50 µg L^−1^ for all toxins. However, if the R^2^ of the linear fit was equal or higher than 0.99, a linear fit was used for quantification.

Furthermore, the matrix effect in the BGAS was determined with an additional method during the validation. For each toxin, a standard curve between 0.1 and 50 µg L^−1^ was measured in the solvent solution (MeOH:H_2_O Milli-Q 50:50 +1% acetic acid) and in the blank matrix (annotated as addition). The slopes of the resulting curves were compared using a student *t*-test, and a matrix effect was deduced when the slopes were significantly different (t(b) > t (95%)).

### 5.6. DNA Extraction, PCR with HEP Primers, and Sequencing of PCR Products

DNA was extracted from nine BGAS samples (Apha-430, Mx-582, Apha-584, Apha-585, Apha-587, Apha-650, Apha-696, Apha-862, Apha-1230), for which contamination with MCs was found by the LC-MS/MS method in this study. DNA extracted from the *Microcystis aeruginosa* strain PCC7806 and *Aphanizomenon gracile* Niva-Cya626 were used as positive controls in the PCR, while DNA from the ‘spirulina’ sample Sp-475, in which no MCs were found in this study, was used as a negative control. Approximately 50 mg of powder or tablets (previously grinded for toxin analysis) were used for DNA extraction with the NucleoSpin Tissue kit (Macherey-Nagel, Düren Germany), following the supplier’s recommendations, with the addition of an initial crushing of the samples with a pestle and glass beads. Thereafter, DNA samples were cleaned with the Genomic DNA Clean & Concentrator kit (Zymo). The DNA samples were stored at −20 °C until further processing. PCR was performed on these DNA samples with the HEP primers HEPF (5′-TTTGGGGTTAACTTTTTTGGGCATAGTC-3′) and HEPR (5′-AATTCTTGAGGCTGTAAATCGGGTTT-3′) amplifying the aminotransferase (AMT) domain situated in the *mcy*E and *nda*F genes encoding the microcystin and nodularin synthetase enzyme complexes [50]. The cycling protocol included an initial denaturation step at 98 °C (1 min), followed by 40 cycles of a denaturation at 98 °C (30 s), an annealing at 57 °C (45 s) and an extension at 72 °C (40 s), and a final extension at 72 °C (7 min). The reaction mix contained in a total volume of 50 µL, 1U Q5 High Fidelity DNA polymerase (New England Biolabs, Ipswich, MA, USA), 1× Q5 Reaction Buffer, 200 μM dNTP, 1 mg mL^−1^ BSA, 0.5 μM of both primers, and 1 µL gDNA. Negative controls without DNA were processed in parallel. The presence or absence of bands of the expected size was visualised after electrophoresis of 4 µL of PCR product on 1.0% agarose gel. PCR products were purified with the NucleoSpin Gel and PCR Clean-up Purification kit (Macherey-Nagel, Düren Germany) according to the furnisher’s protocol, and sent for sequencing with the HEPF primer to the GIGA-GenoTranscriptomics Platform (ULiege, Liège, Belgium). Amplicon sequences were deposited under the GenBank accessions MW924666 to MW924672.

### 5.7. Exposure and Risk Assessment

The exposure to toxins via intake of food supplements should be estimated by using the consumption data for the food supplements. However, these data are scarce. For a more realistic approach, the posology data for each food supplement analyzed in the study were collected. These data were used to estimate the exposure by multiplying the daily dose suggested by the producer with the determined toxin concentration (average and maximum concentrations) to estimate the exposure. The worst case exposure scenario was estimated only for the BGAS where MCs were detected (n = 8). Additionally, a comparison was performed where the consumption data from the EFSA Comprehensive Food Consumption data was used for the same calculations [62].

To estimate the exposure, the intake was calculated based on all analyzed samples (real case) and based on only the concentrations found in the contaminated samples (worst case exposure scenario). For the latter, the values for non-detected were imputed according to the bounding approach. All non-detected values were replaced by LOD values and all not quantified values by LOQ value (upper bound approach). The calculation was performed using the following equation for estimated daily intake (EDI):

Equation (1): formula to calculate estimated daily intake (EDI)
(1)$$EDI\;\left(\frac{\frac{µg}{\mathrm{kg}\;\left(\mathrm{bw}\right)}}{\mathrm{day}}\right)=\frac{concentration_{\frac{µg\left(MC\;total\right)}{g\;\left(BGAS\right)}}\times daily\;dose_{g\;\left(BGAS\right)}}{\mathrm{bw}\;\left(\mathrm{kg}\right)}$$

## Figures and Tables

**Figure 1 toxins-14-00513-f001:**
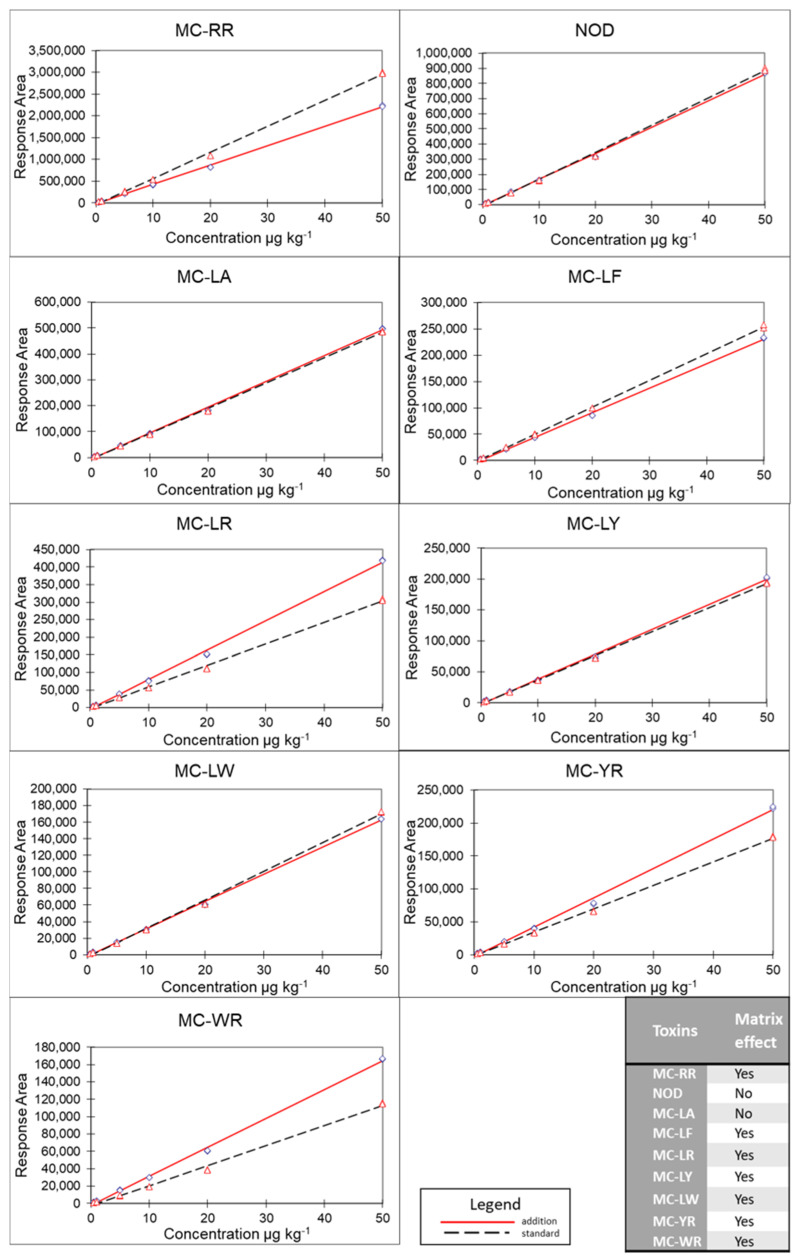
Matrix effect in a blank ‘spirulina’ matrix is assessed during the validation by comparing the response area of a calibration curve in blank matrix (e.g., addition) to a calibration curve in solvent (e.g., standard). Matrix effect is not observed when the curves are parallel to each other. However, when the curves intersect, there is a matrix effect.

**Figure 2 toxins-14-00513-f002:**
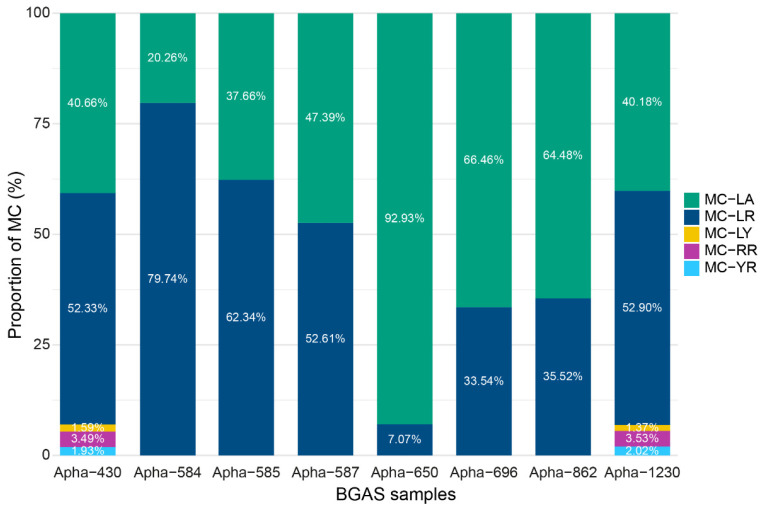
The proportionate contribution (%) of each microcystin congener to the total concentration of microcystin congeners (MCs) per sample of BGAS.

**Table 1 toxins-14-00513-t001:** Overview of results for the validation parameters: recovery, repeatability, reproducibility and measurement uncertainty are calculated for each microcystin congener (MC), Nodularin (NOD) and the sum at each independent concentration level and on average.

Toxins	Spiked Concentration (µg kg^−1^)	Recovery (%)	Repeatability (%)	Reproducibility (%)	Measurement Uncertainty (%)	Average S/N * LOD (22.5 µg kg^−1^)	Average S/N LOQ (50 µg kg^−1^)	R^2^ **
**MC-RR**	50	80.00	4.20	13.26	26.53	224.16	445.87	0.998
	400	76.00	8.87	9.46	18.91			
	800	84.00	1.64	14.02	28.05			
	1200	83.00	1.10	11.68	23.36			
	Average	80.75	3.95	12.11	24.21			
**NOD**	50	95.00	3.30	7.73	15.46	220.22	328.44	0.998
	400	86.00	8.94	15.46	30.92			
	800	89.00	2.50	15.55	31.11			
	1200	89.00	0.91	13.19	26.39			
	Average	89.75	3.91	12.99	25.97			
**MC-LA**	50	102.00	2.79	10.52	21.03	35.10	74.09	0.998
	400	95.00	8.47	13.60	27.20			
	800	99.00	2.06	6.95	13.90			
	1200	99.00	1.17	4.33	8.65			
	Average	98.75	3.62	8.85	17.69			
**MC-LF**	50	102.00	3.72	8.06	16.11	34.01	78.61	0.999
	400	99.00	9.37	10.96	21.91			
	800	103.00	1.40	14.05	28.10			
	1200	103.00	1.48	14.33	28.67			
	Average	101.75	3.99	11.85	23.70			
**MC-LR**	50	92.00	5.61	7.06	14.12	73.46	142.57	0.998
	400	79.00	8.86	18.74	37.48			
	800	82.00	2.54	13.72	27.45			
	1200	82.00	1.88	7.04	14.08			
	Average	83.75	4.72	11.64	23.28			
**MC-LY**	50	102.00	3.50	8.01	16.02	25.53	49.62	0.998
	400	94.00	9.07	15.53	31.06			
	800	96.00	1.21	8.80	17.61			
	1200	96.00	0.75	3.13	6.26			
	Average	97.00	3.63	8.87	17.74			
**MC-LW**	50	103.00	3.77	3.85	7.70	23.62	50.09	0.998
	400	100.00	8.50	11.25	22.50			
	800	104.00	2.24	15.13	30.26			
	1200	103.00	1.64	14.54	29.07			
	Average	102.50	4.04	11.19	22.38			
**MC-YR**	50	86.00	3.31	9.85	19.70	33.86	68.87	0.997
	400	73.00	10.21	16.81	33.61			
	800	75.00	1.85	16.21	32.42			
	1200	75.00	1.93	12.24	24.47			
	Average	77.25	4.33	13.78	27.55			
**MC-WR**	50	90.00	3.24	16.00	32.00	29.17	79.90	0.999
	400	69.00	9.59	15.89	31.78			
	800	69.00	2.04	13.88	27.66			
	1200	69.00	2.01	9.65	19.30			
	Average	74.25	4.22	13.86	27.69			
**Sum**	450	95.00	1.90	4.98	9.95	N.A.	N.A.	N.A.
	3600	86.00	8.98	13.17	26.34			
	7200	89.00	1.62	11.98	34.96			
	10800	89.00	0.83	9.48	30.22			
	Average	89.75	3.33	9.90	25.37			

* Signal to Noise (S/N) is assessed as parameter for the limit of quantification (LOQ). LOQ is assessed at the lowest spiked concentration with an acceptable threshold of 10. S/N is assessed for the limit of detection (LOD) at the lowest concentration in the calibration curve with an acceptable threshold of 3. ** Linearity of the calibration curve for each toxin is determined by R^2^.The LOD and LOQ are assessed for each toxin separately. LOD and LOQ is not calculated for the sum of the concentrations for all toxins and is therefore not applicable (N.A.).

**Table 2 toxins-14-00513-t002:** This table shows the values for the calculated t(b) compared with the tabulated t at the 95% confidence level (2.09) to determine the presence of a matrix effect. If t(b) is higher than t(95%), a matrix effect is present. The t(b) value compares a significant difference between the slope of standard and addition curve.

Toxins	MC-RR	NOD	MC-LA	MC-LF	MC-LR	MC-LY	MC-LW	MC-YR	MC-WR
**t(b)**	19.46	1.76	1.24	7.70	17.53	2.42	3.14	10.45	15.60

**Table 3 toxins-14-00513-t003:** Overview of samples ID with the type of BGAS (pill, powder or tablet), the species indicated on the package, the type of retailer, the total microcystin concentration found, the proposed dose per day and estimated daily consumption.

Sample ID	Type	Species Indicated on the Package	Type of Retailer	Total Microcystin (µg kg^−1^)	Proposed Dose day^−1^ *	Estimated Daily Consumption (g)
**Apha-430**	powder	*Aphanizomenon flos-aquae*	Health food store	5645.33	2 teaspoons or sprinkel	4
**Apha-584**	powder	*Aphanizomenon flos-aquae*	webshop	431.26	1 teaspoon (2 g)	2
**Apha-585**	powder	*Aphanizomenon flos-aquae*	webshop	308.11	1 teaspoon (1.5 g)	1.5
**Apha-587**	capsules	*Aphanizomenon flos-aquae*	webshop	238.45	2 capsules (800 mg/capsule)	1.6
**Apha-650**	powder	*Aphanizomenon flos-aquae*	webshop	1106.06	1 or 2 teaspoons (2 = 3 g)	3
**Apha-696**	tablets	*Aphanizomenon flos-aquae*	webshop	499.52	first 15 days: 2 capsules, after 4 caplsules (400 mg/tablet)	1.6
**Apha-862**	capsules	*Aphanizomenon flos-aquae*	webshop	623.39	2 capsules (800 mg/capsule)	1.6
**Apha-1230**	powder	*Aphanizomenon flos-aquae*	Health food store	5436.31	2 teaspoons or sprinkel	4
**Sp-589**	tablets	*Arthrospira maxima*	webshop	<LOD	3 capsules (600 mg/tablet)	1.8
**Sp-698**	capsules	*Arthrospira maxima*	webshop	<LOD	3 capsules (600 mg/tablet)	1.8
**Sp-433**	tablets	*Arthrospira pacifica*	Health food store	<LOD	6 tablets (500 mg/tablet)	3
**Sp-431**	powder	*Arthrospira platensis*	Health food store	<LOD	Not mentioned on package	2
**Sp-432**	tablets	*Arthrospira platensis*	webshop	<LOD	up to 6 tablets (500 mg/tablet)	3
**Sp-475**	powder	*Arthrospira platensis*	Health food store	<LOD	2 teaspoons (6 g) or sprinkel	6
**Sp-543**	powder	*Arthrospira platensis*	Health food store	<LOD	1 teaspoon (1.5 g)	1.5
**Sp-544**	powder	*Arthrospira platensis*	Health food store	<LOD	1 tablespoon (7 g)	7
**Sp-545**	tablets	*Arthrospira platensis*	Health food store	<LOD	6–10 tablets (300 mg/tablet)	3
**Sp-579**	powder	*Arthrospira platensis*	webshop	<LOD	adults 3 g, children 1.5 g	3
**Sp-581**	powder	*Arthrospira platensis*	webshop	<LOD	Not on package	2
**Sp-583**	powder	*Arthrospira platensis*	webshop	<LOD	2 teaspoons (6 g) or sprinkel	6
**Sp-586**	powder	*Arthrospira platensis*	webshop	<LOD	1 teaspoon (1 g)	1
**Sp-588**	tablets	*Arthrospira platensis*	webshop	<LOD	2–6 tablets (500 mg/tablet)	3
**Sp-651**	powder	*Arthrospira platensis*	webshop	<LOD	1 teaspoon	2
**Sp-863**	tablets	*Arthrospira platensis*	webshop	<LOD	2–3 tablets (500 mg/tablet)	6
**Sp-864**	powder	*Arthrospira platensis*	webshop	<LOD	2 × 1 measuring spoon (±3 g)	6
**Sp-865**	powder	*Arthrospira platensis*	webshop	<LOD	1 teaspoon	2
**Mx-582**	powder	*Arthrospira platensis, Chlorella vulgaris, Aphanizomenon flos-aquae*	webshop	< LOQ	2 teaspoons or sprinkel	4
**Mx-821**	powder	*Arthrospira platensis, Chlorella vulgaris*	webshop	<LOD	1 teaspoon first two weeks than 1 eatspoon (7.5 g)	7.5
**Mx-822**	powder	*Arthrospira platensis, Chlorella vulgaris*	webshop	<LOD	2 × 1 teaspoon	4
**Ch-546**	powder	*Chlorella vulgaris*	Health food store	<LOD	3 teaspoons	6
**Ch-547**	tablets	*Chlorella vulgaris*	Health food store	<LOD	1–12 tablets (500 mg/tablet)	6
**Ch-580**	powder	*Chlorella vulgaris*	webshop	<LOD	Not on package	2
**Ch-649**	powder	*Chlorella vulgaris*	webshop	<LOD	1 spoon (unspecified)	7.5
**Ch-652**	tablets	*Chlorella vulgaris*	webshop	<LOD	6 tablet (500 mg/tablet) 12 in special cases	3
**Ch-697**	tablets	*Chlorella vulgaris*	webshop	<LOD	4 to 6 tablets (400 mg/tablet)	2.4
**Sp-823**	powder	*Spirulina platensis/maxima*	webshop	<LOD	Not mentioned on package	2

* The weight of BGAS measured as a teaspoon suggested by EFSA (2 g), was used when the proposed daily dose is not mentioned on the package or the weight of a teaspoon is not mentioned.

**Table 4 toxins-14-00513-t004:** Mean, median and 95th percentile estimated daily intake (EDI) (µg kg_bw_^−1^ day^−1^) for an adult population to microcystins detected in BGAS.

Exposure Scenario	EDI µg kg_bw_^−1^ day^−1^		
** *adults (consumption following the posology recommendation)* **	**mean**	**median**	**P95**
real (UB) *	0.02	0.00	0.14
worst (UB) **	0.10	0.02	0.33
** *adults (mean consumption as calculated from the EFSA database* ** ******)***	**mean**	**median**	**P95**
real (UB) *	0.01	0.00	0.06
worst (UB) **	0.04	0.01	0.13

* Real (UB), the exposure scenario where all analyzed BGAS were taken into account (upper bound approach); ** worst (UB), only results of the positive samples (at least one MC detected) were taken into account (upper bound approach), *** EFSA, the mean chronic consumption was calculated from the EFSA Food Consumption Comprehensive Database.

**Table 5 toxins-14-00513-t005:** MS/MS parameters used for the ion fragmentation.

Toxins	Precursor Ion (*m/z*)	Quantifier Ion (*m/z*)	Collision Energy (eV)	Cone Voltage (V)	Qualifier Ion (*m/z*)	Collision Energy (eV)	Cone Voltage (V)
**MC-LR**	995.4	135.0	70	80	213.1	60	80
**MC-RR**	519.8	134.8	30	50	107.2	60	50
**MC-YR**	1045.5	135.3	80	60	212.9	60	60
**MC-WR**	1068.4	135.3	70	100	213.1	60	100
**MC-LY**	1002.4	135.4	60	50	213.0	50	50
**MC-LA**	910.3	135.1	60	50	107.1	80	50
**MC-LF**	986.3	135.0	60	70	213.1	60	70
**MC-LW**	1025.4	134.9	60	60	213.1	50	60
**NOD**	825.25	134.9	50	80	102.7	90	80

## Data Availability

Not applicable.

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
