# Peer review of "LC-MS/MS Validation and Quantification of Cyanotoxins in Algal Food Supplements from the Belgium Market and Their Molecular Origins"

_toxins, 2022, doi:10.3390/toxins14080513_

Round 1

Reviewer 1 Report

Dear authors and editors,

The manuscript LC-MS/MS validation and quantification of cyanotoxins in algal food supplements and their molecular origins presents valuable data on the validation of a method based on Ultra High Performance Liquid Chromatography-Tandem Mass Spectrometry to quantify microcystin congeners and nodularin in algal food supplements. In order to improve the final version of the manuscript, I have some notes and comments:

General

1. Given that in the body of the text authors comments many times Belgium markets, in my opinion the title of the proposed manuscript should be changed and supplemented with the origin of the food supplements: for example,.….algal food supplements in Belgium market and …..

2. It is not clear why spirulina is written in lowercase and normal font, while according to the International Code of Nomenclature for algae, fungi, and plants the Latin names of the algal genera should be capitalized and italicized. I guess the authors probably used spirulina to mark the market product, but it is not the same with Chorella (line 30). In my opinion, it is better to follow the International Code because the specific product from the market actually contains and is related to a representative of some algal genus or species. Please correct this throughout the text or make an arrangement at the very beginning for the writing of the name of the algal products.

Minor

·       on line 37 is written phyto-, myco-, or phytotoxins. Most probably it should be phyco- myco-, or phytotoxins or only myco-, or phytotoxins

·       on line 50 in the Latin name Arthrospira sp.  – sp. should not be italicized

·       on line 64 is written hepatoxins instead of hepatotoxins

·       on line 72 is written Oscillaria instead of Oscillatoria

·       in Table 2 all algal species names should be italicized

·       on line 286-287 Aphanizomenon flos-aquae should be italicized

Reviewer 2 Report

The manuscript on determination of toxins in food supplements is very interesting, and worth to be published after some improvements, as shown below:

1) English can be polished, e.g., “the presence of toxin-producing cyanobacteria” – I think that authors mean “the presence of toxin produced by cyanobacteria”.

2) I think that all statistic data could be shown in the paper (e.g., as supporting file), including also results for Student's t-test.

3) The colors in Figure 2 could be better selected – now it is difficult to see differences between three green types.

4) Authors could write more sound conclusions. I wonder if in their opinion, it is better to take or not the food supplements containing cyanotoxin. Do these food supplements have more positive than negative impact on our health?  

5) I do not understand some sentences/phrases, e.g.,

“However, similar consumption data is not yet available for Europe nor Belgium, which prevents more realistic assessment of a health risk.” It is unclear what kind of consumption data were published in reference [2] – if not for Europe? Why is particularly Belgium mentioned here – is it some special case for food supplement use? If not, I think no need to underline Belgium here and in Abstract. If this is important (next sentence), then it should be clarified why authors focus on Belgium.

6) General advice about scientific writing:

i)                    Citations should be avoided in the abstract.

ii)                   The use of future tenses is also not recommended (in except of planned experiments for the future study), e.g., “Cylindrospermopsin is an alkaloid with a cyclic guanidine that will inhibit, when ingested, protein synthesis in the cell and thus cell growth..”

Reviewer 3 Report

I am not sure what is the highlight of the paper, as the author stated that LC-MS(/MS) has evolved to become a commonly used technique to detect MCs in BGAS. If the purpose of this paper is to emphasize the advantages of PCR amplification of the MCYE gene, the discussion should focus on it. As for PCR amplification of the MCYE gene, what is the detection limit? What are the advantages of LC-MS/MS compared to LC-MS? Better to compare and discuss again.

Specific comments

l  The introduction is too long and does not highlight the significance of the paper.

l  In line 43, 50, Arthrospira sp. should be Arthrospira sp.

l  In Table 1, the meaning of Repeatability (%) Reproducibility (%) is not clear, what do they means?

l  In line 181, Microcystis sp. should be Microcystis sp.

Round 2

Reviewer 3 Report

After revision, the manuscript has been greatly improved. However, there are still some issues to be resolved. Figure 1 is not clear enough, which should be re-drawn. More importantly, the author does not give a clear answer about the detection limit of MCYE gene amplification.
